# Perceptual Decision Efficiency Is Modifiable and Associated with Decreased Musculoskeletal Injury Risk Among Female College Soccer Players

**DOI:** 10.3390/brainsci15070721

**Published:** 2025-07-04

**Authors:** Gary B. Wilkerson, Alejandra J. Gullion, Katarina L. McMahan, Lauren T. Brooks, Marisa A. Colston, Lynette M. Carlson, Jennifer A. Hogg, Shellie N. Acocello

**Affiliations:** 1Department of Health & Human Performance, University of Tennessee at Chattanooga, Chattanooga, TN 37403, USA; marisa-colston@utc.edu (M.A.C.); lynette-carlson@utc.edu (L.M.C.); jennifer-hogg@utc.edu (J.A.H.); shellie-acocello@utc.edu (S.N.A.); 2Graduate Athletic Training Education Program, University of Tennessee at Chattanooga, Chattanooga, TN 37403, USA; gmn283@mocs.utc.edu (A.J.G.); tlt381@mocs.utc.edu (K.L.M.); zmz916@mocs.utc.edu (L.T.B.)

**Keywords:** virtual reality, perceptual decision-making, performance enhancement, injury prevention, cognitive training

## Abstract

**Background:** Prevention and clinical management of musculoskeletal injuries have historically focused on the assessment and training of modifiable physical factors, but perceptual decision-making has only recently been recognized as a potentially important capability. Immersive virtual reality (VR) systems can measure the speed, accuracy, and consistency of body movements corresponding to stimulus–response instructions for the completion of a forced-choice task. **Methods**: A cohort of 26 female college soccer players (age 19.5 ± 1.3 years) included 10 players who participated in a baseline assessment, 10 perceptual-response training (PRT) sessions, a post-training assessment that preceded the first soccer practice, and a post-season assessment. The remaining 16 players completed an assessment prior to the team’s first pre-season practice session, and a post-season assessment. The assessments and training sessions involved left- or right-directed neck rotation, arm reach, and step-lunge reactions to 40 presentations of different types of horizontally moving visual stimuli. The PRT program included 4 levels of difficulty created by changes in initial stimulus location, addition of distractor stimuli, and increased movement speed, with ≥90% response accuracy used as the criterion for training progression. Perceptual latency (PL) was defined as the time elapsed from stimulus appearance to initiation of neck rotation toward a peripheral virtual target. The speed–accuracy tradeoff was represented by Rate Correct per Second (RCS) of PL, and inconsistency across trials derived from their standard deviation for PL was represented by intra-individual variability (IIV). Perceptual Decision Efficiency (PDE) represented the ratio of RCS to IIV, which provided a single value representing speed, accuracy, and consistency. Statistical procedures included the bivariate correlation between RCS and IIV, dependent *t*-test comparisons of pre- and post-training metrics, repeated measures analysis of variance for group X session pre- to post-season comparisons, receiver operating characteristic analysis, and Kaplan–Meier time to injury event analysis. **Results:** Statistically significant (*p* < 0.05) results were found for pre- to post-training change, and pre-season to post-season group differences, for RCS, IIV, and PDE. An inverse logarithmic relationship was found between RCS and IIV (Spearman’s Rho = −0.795). The best discriminator between injured and non-injured statuses was PDE ≤ 21.6 (93% Sensitivity; 42% Specificity; OR = 9.29). **Conclusions:** The 10-session PRT program produced significant improvement in perceptual decision-making that appears to provide a transfer benefit, as the PDE metric provided good prospective prediction of musculoskeletal injury.

## 1. Introduction

Competitive advantages in sports and the avoidance of sport-related injury are widely recognized to be heavily dependent on physical performance capabilities, including muscular strength, power, endurance, speed, and flexibility, but the neural efficiency of cognitive processes has not historically been recognized as a fundamental factor that is modifiable [1,2,3,4]. Reaction time, or response time (RT), is widely viewed as a valid behavioral indicator of brain processing speed [5], but rapid muscle responses to sensory inputs only have value when they contribute to effective (i.e., correct) goal-directed actions [6,7]. Perceptual decision-making is defined as the process of discriminating between sensory stimuli for selection of a response among alternative options [8], which represents a higher order cognitive process than the generation of simple reactive responses to environmental stimuli. In fact, sport performance can be conceptualized as a series of perceptual decisions that are made under rapidly changing environmental conditions [9], and the impairment of perceptual decision-making appears to increase injury susceptibility [10]. Given the importance of cognitive processes, the practical challenges to realize the potential benefits of enhanced neural efficiency is the identification of the best performance metrics for its assessment and a training method that will produce beneficial adaptations [11,12,13].

Virtual reality (VR) provides a means to control the timing, movement patterns, and forms of successive visual stimuli presented within an immersive head-mounted display, and to precisely quantify the amount of time that elapses until the expression of a given decision through the initiation of a specified stimulus–response movement [14]. Ideally, the design of testing and training procedures should reflect current understanding of neural mechanisms associated with perceptual decision-making [15,16]. Although brain–behavior relationships are exceedingly complex, numerous studies have provided evidence that the drift–diffusion computational model of decision-making provides a valid representation of the distinct neural processes of sensory evidence accumulation and adjustments in a decision threshold [7,17,18,19,20,21,22,23,24,25,26]. Moment-to-moment variation in the signal-to-noise ratio of accumulating evidence (i.e., diffusion) is transmitted from the middle temporal area to the lateral intraparietal sulcus, which drives its rate of neuronal firing toward a decision threshold (i.e., drift). A separate neural circuit involving the medial prefrontal cortex and the subthalamic nucleus monitors the amount of evidence needed to avoid an incorrect decision, with an upward or downward adjustment of the decision threshold to optimize the speed–accuracy tradeoff for subsequent decisions [20]. Behavioral manifestations of these neural processes can be approximated by metrics derived from a two-alternative, forced-choice task [27], which may be optimized within an immersive VR environment.

Perceptual latency (PL) has previously been defined as the time that elapses from visual stimulus appearance to the earliest detectable initiation of a movement response (i.e., six degrees of angular displacement of the neck toward a peripherally located response target), which estimates the duration of the perceptual decision-making process and excludes the time required for full execution of a movement response [28]. The intra-individual variability (IIV) of the elapsed decision time across trials (i.e., trial-to-trial inconsistency) is believed to be an indicator of a suboptimal signal-to-noise ratio [26]. An elevated IIV and a low signal-to-noise ratio are believed to result from interrelationships among intrinsic neural dynamics [8,29,30,31,32], stochastic neuronal spiking activity [7,33,34], attentional lapses [35,36,37,38,39], microstructural disruption of white matter tracts [40,41,42], fluctuations in synchronization of neural signals [43,44,45], or alterations in functional connectivity between and within brain networks [46,47,48]. The findings of previous research suggest that IIV may be viewed as a behavioral efficiency index to represent an individual’s perceptual decision-making capability [38]. Thus, a high IIV across successive trials would be an indicator of low neural efficiency [34].

The Rate Correct per Second (RCS) of the elapsed perceptual decision-making time for all trials provides a VR metric representing the speed–accuracy tradeoff with a single value [49], which reflects the neural mechanism responsible for setting the decision threshold [22,25]. Although drift–diffusion evidence accumulation and decision threshold adjustments are distinct processes, high-quality sensory inputs could be expected to produce a faster drift rate and fewer incorrect decisions, and interactions across multiple decisions made in close succession could be expected to produce an inverse correlation between the respective behavioral manifestations represented by RCS and IIV [50]. If so, a Perceptual Decision Efficiency (PDE) metric derived from the ratio of RCS to IIV for neck PL would provide a single value representing decision speed, accuracy, and consistency.

Some research evidence has demonstrated that VR training can improve behavioral metrics associated with neurophysiological mechanisms underlying perceptual decision-making [51,52,53], but relatively little evidence of the transfer to real-world benefits from such training currently exists [13]. In principle, a perceptual decision-making task that progressively becomes more difficult should elicit neuroplastic adaptations within neural circuits that improve performance capabilities [1,2,54]. Arguably, the most important transfer effect for athletes would be a reduction in injury incidence [55]. Female soccer players are among athletes with the greatest level of exposure to injury risk, which may be increased by repetitive sub-concussive head impacts associated with heading or their combined effect with a history of concussion [56,57,58]. Accordingly, the purposes of this study were to: (1) assess the magnitude of improvement in neck PL performance metrics following a total of 10 VR perceptual response training sessions completed within a 10-day period prior to the initiation of pre-season practice sessions for a group of 10 college women’s soccer players; (2) compare the incidence of musculoskeletal sprain, strain, or overuse injury among a cohort of 26 college women’s soccer players (10 trained and 16 untrained) over a 14-week period of practice sessions and games; and (3) evaluate the relationships among VR metrics that may advance the understanding of brain processes controlling observable goal-directed behavior. The novel aspect of this study is the derivation of a composite VR metric that is expected to be prospectively associated with the likelihood for musculoskeletal injury occurrence.

## 2. Materials and Methods

### 2.1. Participants and Institutional Review Board Statement

A total of 26 college women’s soccer players (19.5 ± 1.3 years, 1.68 ± 0.06 m, 63.57 ± 6.75 kg), who comprised the entire roster of an NCAA Division I team, participated in the study. As members of a collegiate team roster, each athlete provided consent and authorization for access to protected health information for research purposes. All procedures were approved by the Institutional Review Board of the University of Tennessee at Chattanooga (#23-052 approved on 8 May 2023). The study complied with the Declaration of Helsinki (DoH)—Ethical Principles for Medical Research Involving Human Participants (1964) and its latest amendments adopted by the 75th General Assembly of the World Medical Association (WMA) in Finland on 19 October 2024. The only exclusionary criterion was an injury-related impairment that limited the ability to perform rapid arm reaching and single-step lunging movements.

### 2.2. Procedures

Prior to participation in a baseline VR test, each player completed an electronic version of the Global Well-Being Index [59], which recorded introspective ratings of their current statuses pertaining to: (1) general pain/discomfort (headaches, neck pain, non-specific body discomfort), (2) sleep-related problems (trouble falling asleep, sleeping less, fatigue/drowsiness), (3) mood-related problems (nervousness/anxiety, sadness/depression, irritability/stress), (4) musculoskeletal problems during activities of daily living (aching discomfort, joint stiffness, muscle spasms/tightness), and (5) high-intensity performance limitations (running speed, explosive power, endurance). A positive response within a category triggered follow-up queries to rate frequency, temporal recency, and severity. The sum of the 0–10 values for each of the five categories yielded a 0–50 introspective dysfunction rating (i.e., a high score indicating suboptimal well-being). Self-reported lifetime histories of concussion and musculoskeletal sprains and strains sustained during the previous 12-month period were also documented by the selection of response options on the electronic survey administered through the REDCap (Research Electronic Data Capture) system [60]. All musculoskeletal injuries that were evaluated and treated by athletic trainers over the course of a 14-week period, which included all pre-season practice sessions, in-season practice sessions, and a total of 19 games, were electronically documented.

A convenience cohort of 10 players who were physically present two weeks prior to the start of the pre-season practice sessions participated in a baseline VR test of perceptual decision-making, followed by 10 perceptual response training (PRT) sessions scheduled on consecutive days. Each VR session (i.e., testing and training) involved 40 trials of neck rotation, arm reaching, and single-step lower extremity lunging movements in response to light blue visual stimuli that moved horizontally across the black background of the head-mounted display. The baseline test corresponded to the initial difficulty level for PRT, which presented a single stimulus per trial that indicated the correct left or right directional response for hand controller movement to a peripherally located virtual target: (1) a horizontally moving circle dictated a movement response in the same direction as that of the circle, whereas (2) a horizontally moving ring dictated a movement response in the direction opposite that of the ring.

A transition to a greater level of task difficulty was made for the next PRT session that followed a correct response rate of 90 percent or greater (≥36/40). Each of the three progressively higher difficulty levels included two distractor stimuli (circles, rings, or one of each). The two highest difficulty levels introduced variability in the initial location of the target stimulus in relation to the distractor stimuli, and the highest difficulty level increased the movement speed of the stimuli. The 10 players who completed the PRT program performed a follow-up VR test that corresponded to the difficulty level of the baseline test on the day after the tenth training session. The remaining 16 players who did not train completed the baseline VR test 1–3 days prior to the first pre-season practice session. Thus, the follow-up tests for the 10 training group (TG) players and the baseline tests for the 16 untrained comparison group (CG) players were designated as pre-season tests, which were compared to post-season tests administered at the end of the 14-week period of practice sessions and games. Figure 1 presents a flowchart that depicts the sequence of VR testing and training.

The immersive VR system (PICO Neo3 Pro Eye, PICO Immersive, Ltd., Mountain View, CA, USA) quantified the amount of time that elapsed from the stimulus appearance to the initiation of movement toward a peripheral virtual target (i.e., PL) for eye displacement (6 degrees), neck rotation (6 degrees), arm reaching with the hand controller (10 cm), and lower extremity single-step lunging (10 cm). Response time (RT) for the same movements quantified the amount of time elapsed from the stimulus presentation to the maximum extent of horizontal displacement toward the virtual target (Figure 2). Further details of the immersive VR testing procedures and the rationale for the derivation of the various performance metrics have been presented in previous reports [28,52,53,59,61]. The prospective associations with musculoskeletal injury have been documented for arm RT [59], as well as arm RCS-RT and arm IIV-RT [53]. The VR metric that demonstrated the strongest retrospective association with concussion history among college wrestlers was neck IIV-PL [52], which may provide a more isolated representation of cognitive function than arm IIV-RT. Among the various VR metrics, test–retest reliability was greatest for neck mean-PL (ICC = 0.922) and neck IIV-PL (ICC = 0.836) over three measurement sessions on consecutive days [28]. The same dataset was subsequently used to calculate values for neck RCS-PL (ICC = 0.916) and neck PDE (ICC = 0.842).

### 2.3. Data Analysis

The predictive modeling of the potential benefit of immersive VR training for injury prevention was the focus of this study, rather than the testing of specific hypotheses. Accordingly, the statistical significance of a given test result was based on α < 0.05, without adjustments for multiple comparisons. Group comparisons for potentially confounding factors utilized chi-square tests for count data and Fischer’s exact test for binary categorizations. A Mann–Whitney test was used to compare 0–50 introspective dysfunction ratings between groups.

The Shapiro–Wilk test was used to assess the distribution normality of each neck PL metric for each measurement session, and natural log transformation was applied to improve the normality of positively skewed data. Original and log-transformed data were analyzed by dependent *t*-tests for TG pre- and post-training comparisons, as well as group X session repeated measures analysis of variance for pre- and post-season assessments. Cohen’s d quantified the effect size associated with the dependent *t*-test results (≥0.2 small, ≥0.5 medium, ≥0.8 large), whereas partial eta-squared quantified the effect size for the group X session results (≥0.01 small, ≥0.06 medium, ≥0.14 large) [62].

Bivariate regression curve estimation was used to evaluate an expected relationship between IIV and RCS. Receiver operating characteristic analysis was used to identify an optimal cut point for the binary classification of musculoskeletal injury risk, which was evaluated for its discriminatory value by calculation of its sensitivity, specificity, and odds ratio (OR). The magnitude of the OR (≥1.32 small, ≥2.38 medium, ≥4.70 large) was used to qualitatively assess the effect of the binary risk classification [63]. The cut point classifier found to provide the best discrimination was subsequently used in a Kaplan–Meier time to event analysis. All analyses were performed using SPSS Version 29.0 (Armonk, NY, USA).

## 3. Results

The median age of the TG was 18 years (range 18–21) compared with 20 years for the CG (range 18–22), which was a statistically significant age difference (*p* = 0.024). No significant differences were found for lifetime history of concussion (*p* = 0.701), musculoskeletal injury in the prior year (*p* = 0.422), attention deficit hyperactivity disorder (*p* = 0.508), learning disability (*p* > 0.999), number of games as a starter (*p* = 0.468), or number of games played (*p* = 0.610). Neither was there any significant difference between the 0–50 introspective dysfunction rating between the groups (*p* = 0.938), nor for any of the 0–10 sub-scores for the five problem categories of the Global Well-Being Index.

To ensure that a total of 10 training sessions were completed by each TG player, those who were unavailable on a given day due to schedule conflicts participated in two consecutive 40-trial sessions on the following day. An unknown technology malfunction resulted in the corruption of data for one participant’s post-training assessment, which was imputed with TG mean values for each post-training VR metric. A knee injury sustained by one CG player precluded performance of the post-season assessment. The missing data was imputed with CG mean values for each post-season VR metric.

Natural log (Log_e_) transformation marginally improved the distribution normality of the neck PL metrics for majority of the test sessions. Results are provided for both the original (untransformed) data and Log_e_ data analyzed by dependent *t*-tests for TG pre- and post-training comparisons (Table 1 and Table 2), as well as group (TG versus CG) X session (pre- versus post-season) repeated measures analysis of variance procedures (Table 3, Table 4 and Table 5). The results for the analyses of Log_e_ data present geometric means (i.e., back-transformations of Log_e_ values approximating median values for pre- and post-training data and group X session data). Post hoc power values for the original and Log_e_-transformed data group differences were both 1.000 for Perceptual Decision Efficiency, both 0.998 for Rate Correct per Second, and 0.997 (original data) and 1.000 (transformed data) for Intra-Individual Variability.

Figure 3A–C presents graphic depictions of the changes in neck PL metrics across all testing sessions that include labels for the original untransformed mean values. Figure 4 illustrates an inverse logarithmic relationship between RCS-PL and IIV-PL (Spearman’s Rho = −0.795; *p* < 0.001), which supports the validity of the PDE metric as a highly informative indicator of perceptual decision-making performance. Among the TG players, a change in the ratio of neck mean-PL to arm mean-RT from 0.49 to 0.46 indicates a reduction in the duration of the perceptual decision-making component of the time that elapsed from stimulus appearance to maximum arm displacement toward the virtual target. Arm mean movement time (mean-MT = mean-RT − mean-PL) improved by 20% (0.727 s to 0.580 s), whereas neck mean-PL improved by 30% (0.705 to 0.493 s).

A total of 21 injuries were sustained by 14 players (i.e., 2 players sustained 3 injuries, 3 players sustained 2 injuries, and 9 players sustained a single injury), each of which involved the core or lower extremity: 3 low back, 3 groin/hip, 3 thigh, 6 knee, and 6 lower leg injuries. Group membership did not demonstrate a strong association with injury incidence, which was 63% (10/16) among CG players compared with 40% (4/10) among TG players (OR = 2.50). However, receiver operating characteristic analysis identified pre-season values for neck RCS-PL ≤ 1.64 (79% Sensitivity; 58% Specificity; OR = 6.67) and neck IIV-PL ≥ 0.134 (79% Sensitivity; 50% Specificity; OR = 3.67) as better predictors of a core or lower extremity injury. The best discriminator between injured versus non-injured status was PDE ≤ 21.6 (93% Sensitivity; 42% Specificity; OR = 9.29). Among players who exhibited a high pre-season level of PDE (>21.6), 100% (6/6) were in the TG, and their negative predictive value (i.e., injury avoidance) was 83% (5/6). Figure 5 presents a Kaplan–Meier time to event comparison of cumulative injury occurrences for 20 players with a pre-season PDE of ≤ 21.6 versus the 6 players with a pre-season PDE of > 21.6 (Mantel–Cox Log Rank *p* = 0.059).

## 4. Discussion

### 4.1. Interpretation of the Study Findings

Each of this study’s specified purposes were achieved. To the extent that behavioral measurements of speed, accuracy, and consistency of perceptual responses provide valid estimates of specific brain processes, our findings may represent a substantial advance in the current understanding of brain–behavior relationships [9,21]. The evidence provided by this study, combined with that derived from previous studies that we have cited, strongly suggest that the training-induced adaptations in goal-directed behavioral performance correspond to favorable neuroplastic changes in the brain circuits involved in perceptual decision-making. Because prolonged or incorrect perceptual decision-making responses have been associated with injury occurrences [10], the performance improvements observed after the completion of the 10-session PRT program suggest that this cognitive injury risk factor is modifiable. Although we did not acquire neuroimaging or electrophysiological measurements of brain signals, the theoretical links between our RCS and IIV behavioral metrics and specific elements of the well-validated drift–diffusion model of perceptual decision-making appear to provide a remarkable degree of explanatory detail about the nature of brain–behavior relationships [23,64].

Two key characteristics of brain processes underlying perceptual decision-making that are not widely recognized outside the neuroscience discipline include the following: (1) The presentation of identical sensory stimuli do not generate exactly the same responses from moment to moment [65]. (2) Although the neural activity involved in movement preparation occurs at the same time as the neural activity associated with the accumulation of decision-related sensory evidence [9], movement is not initiated until after an adjustable threshold has been crossed and a decision has been made [21,66]. Both of these phenomena are influenced by constant fluctuations of transient brain states [32,67,68], which correspond to different functional connectivity patterns among key brain networks [29]. The default mode network (DMN), which includes the medial prefrontal cortex and posterior cingulate cortex, is active during periods of inattentiveness to the external world and a relatively greater internal focus of the mental state [69]. The salience network (SN), which includes the anterior insula and dorsal anterior cingulate cortex, maintains functional connectivity with the DMN during an inattentive state, whereas a transition to functional connectivity of the SN with the frontoparietal network (FPN) establishes a task-optimal state that facilitates sensory evidence accumulation and reduces the IIV of responses to environmental stimuli [69].

Inefficient decoupling of the SN from the DMN is believed to create attention lapses that adversely affect both within-trial sensory evidence accumulation and the consistency (i.e., low IIV) of responses across multiple trials [17,36,38]. Depending on somewhat differing definitions in the literature, the FPN is synonymous with, or closely matches, both the executive control network and the dorsal attention network (DAN). Key components of the FPN include the dorsolateral prefrontal cortex and the posterior parietal cortex, which are structurally connected by the dorsal branch of the superior longitudinal fasciculus (SLF-1). The SLF-1 includes white matter projections to specific components of the DAN that include the lateral intraparietal sulcus, the frontal eye field, and the premotor cortex, which each have roles in both sensory evidence accumulation and motor preparation [18]. The SN is synonymous with, or closely matches, the ventral attention network (VAN) that couples with the DAN and suppresses the DMN when activated [29]. Both structural and functional magnetic resonance imaging acquired from the same individuals have demonstrated that SLF-1 volume is associated with the rate of sensory evidence accumulation (i.e., drift rate) and that functional connectivity of the premotor cortex within the DAN is associated with more efficient transformation of sensory inputs into motor responses [18].

To convert our behavioral data into components that best align with those of the drift–diffusion model, a criterion is needed to separate the decision-making process from the execution of the chosen directional neuromuscular response. Because a perceptual-motor response is not initiated until after a decision has been made [21], the onset of eye, neck, arm, or whole-body movement can be used to define the end of the perceptual decision-making process [70,71]. Extremely rapid multidirectional eye movements are required to detect moving visual stimuli of different types and different locations across the visual field of view, whereas body segments exhibit unidirectional movements toward a chosen response target [72]. Hand controller sensors provide a precise quantification of the location and speed of the arm-reaching response endpoint, but the hand controller position between trial initiation and the onset of a chosen response can be highly variable. The need to maintain the head in a forward-facing neutral position for optimal visualization of both the left and right peripheral fields on the VR headset display provides a highly reproducible starting position of the neck, which is followed by a required unidirectional rotational movement of the neck to visualize a virtual response target located beyond the peripheral limit of the initial field of view. Sensors within the VR headset track the angular position within the transverse plane and the speed of angular displacement, which demonstrate a sudden increase in the rate of neck rotation beyond six degrees. Thus, subtraction of the total perceptual-motor response time (i.e., visual stimulus appearance to hand controller response endpoint) from PL (i.e., visual stimulus appearance to six degrees of neck rotation) may provide the most precise estimate of movement time (MT) for motor response execution. The comparison of metrics derived from these PL and MT intervals may be found beneficial to distinguish between perceptual versus motor impairment as a primary source of dysfunction [16].

The use of neck PL to estimate the duration of the perceptual decision-making process does not exclude the amount of time required for encoding and transmitting a neural signal from the retina to the lateral intraparietal sulcus for the initiation of the evidence accumulation process, which is estimated to require approximately 200 ms [25,67]. Additionally, the amount of time that elapses between the end of the perceptual decision-making process and the generation of the efferent motor output has been estimated to be approximately 80 ms [73], but the duration of this interval is likely to vary on the basis of pre-motor movement planning efficiency [66,72]. Non-decision time derived from the analyses of electroencephalography signals or directly recorded neuronal activity can clearly provide precise measurements of perceptual decision time, which is overestimated by behavioral data [26]. Despite this limitation, the RCS and IIV values calculated for the behaviorally estimated neck PL interval demonstrated a strong natural logarithmic correlation that suggests they provide a highly meaningful representation of distinct and interrelated neural processes underlying perceptual decision-making. Numerous aspects of brain function exhibit natural logarithmic organization, including the range of axon diameters, circuit synaptic strengths, neurotransmitter concentrations, and the power spectral density of neuronal signal frequencies [65]. Thus, a strong theoretical rationale supports our development of the PDE metric as a representation of the combination of perceptual response speed and accuracy (i.e., RCS) with performance inconsistency across trials (i.e., IIV).

Large magnitudes of post-training improvements in PDE and its constituent RCS and IIV components were observed, along with substantial retention over a 14-week period that did not include any training sessions. An important consideration for the investment of time and resources in any type of training program is evidence supporting its transfer effect to real-world benefit. We found pre-season PDE was a strong prospective predictor of core or lower extremity occurrence, which suggests that the PRT program reduced susceptibility. The median pre-training PDE value for the 10 players in the TG was 3.00 and the maximum was 12.90. The median pre-season PDE value for the 16 players in the CG was 2.86 and the maximum was 15.26. Counterfactual reasoning supports the idea that fewer injuries might have been sustained among CG players if they had participated in the pre-season PRT program, and that injury occurrences might have been greater among the 6 players who demonstrated a pre-season PDE ≥ 21.6, each of whom were in the TG, if they had not received the benefit of training. Consistent with the view of elevated IIV as a behavioral manifestation of brain dysfunction [17], and reduced IIV as an early marker of training benefit [74], the natural log-transformed IIV metric demonstrated the largest pre- to post-training effect size.

### 4.2. Limitations

The primary limitation of this study was a lack of neurophysiological measurements of perceptual-motor processes that could provide direct evidence of correspondence with our behavioral metrics. Other limitations include its modest cohort number and lack of randomized assignment of participants to experimental and control groups, both of which precluded rigorous control of potentially confounding factors. For example, one of the untrained players demonstrated the second-highest value for Rate Correct per Second (Figure 4), which is presumably attributable to either an innate perceptual capability or some prior experience that developed a favorable adaptation not demonstrated by the other untrained players. Relationships between the VR metrics, responsiveness to the VR training program, and the association of the Perceptual Decision Efficiency metric with musculoskeletal injury incidence may differ for other sex-, age-, and sport-specific athlete populations.

### 4.3. Future Research

Acute and chronic stress are known to interact with mood disorders, sleep disruption, and mild traumatic brain injury to adversely affect autonomic, immune, endocrine, and metabolic processes that can interfere with perceptual decision-making [75]. The possible cumulative effects of multiple sport-related concussions and repetitive sub-concussive head impacts are a major concern for the long-term brain health of soccer players [57,58,76], which have also been documented to elevate risk for musculoskeletal injuries [77]. Despite the resolution of post-concussion clinical symptoms, the ability to simultaneously extract goal-relevant sensory information and to suppress irrelevant environmental stimuli can be a persistent impairment that is not detected by standard clinical assessments [78]. Regardless of sport-related concussion history, brief periods of activity that impose complex perceptual-motor demands can result in a breakdown in decision-related sensorimotor integration [10,79]. More research is needed to better develop resilience to such breakdowns, which may result from the improved integration of long-term memory with sensory inputs and motor planning [11].

Skilled sport performance results from a series of perceptual decisions involving the extraction of relevant sensory information that is integrated with the planning and execution of effective motor responses as a sequence of events rapidly unfold [72]. Increasingly, training for the optimization of perceptual-motor performance is becoming recognized as an important contributor to both enhanced performance capabilities and injury resistance [4,13,80]. The design of an appropriate training activity should incorporate both perceptual challenges and movement responses to promote neuroplastic modifications in the bidirectional circuits transmitting information between decision-making and motor planning processes, which can produce faster movement responses and maximize transfer to performance in a competitive sport environment [12,71,81]. Accordingly, the immersive VR task that we used was not designed to maximize the realism of a specific sport simulation (i.e., physical fidelity), which can be ineffective or counterproductive for behavioral adaptation [82]. The utilization of an immersive task that incorporated the basic elements of the neurophysiological mechanisms underlying perceptual-motor performance (i.e., psychological fidelity) provides the most likely explanation for the results we observed among the female soccer players who participated in the PRT program [16]. Future research might assess far-transfer of training benefit through the comparison of post-training VR metrics to coaches’ ratings of on-field performance during competitive events.

## 5. Conclusions

The findings of this study support the use of the PDE metric as a single value for the representation of a combination of perceptual decision speed, accuracy, and consistency, each of which is clearly modifiable through appropriately designed cognitive–motor training activities. Furthermore, counterfactual reasoning suggests that the 10-session PRT program we administered through immersive VR provided a beneficial reduction in the risk of musculoskeletal injury.

## Figures and Tables

**Figure 1 brainsci-15-00721-f001:**
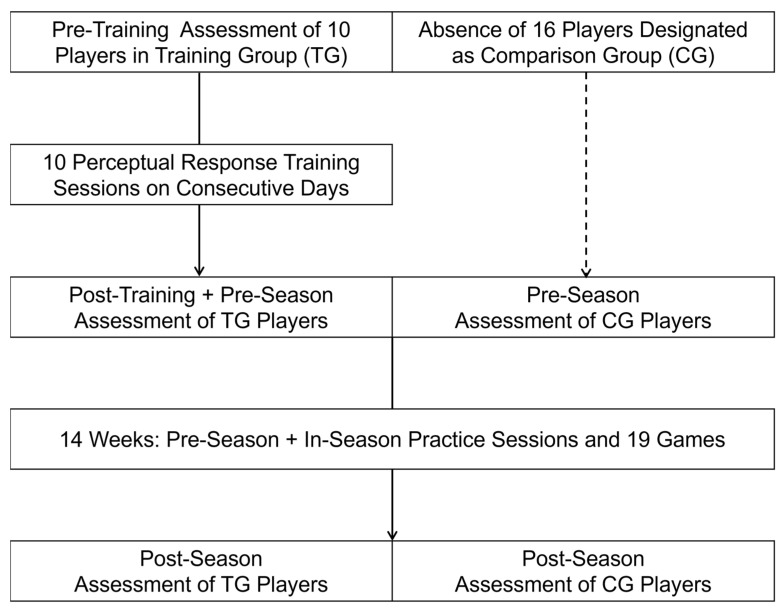
Flowchart depicting the time course of assessments, training, and injury surveillance.

**Figure 2 brainsci-15-00721-f002:**
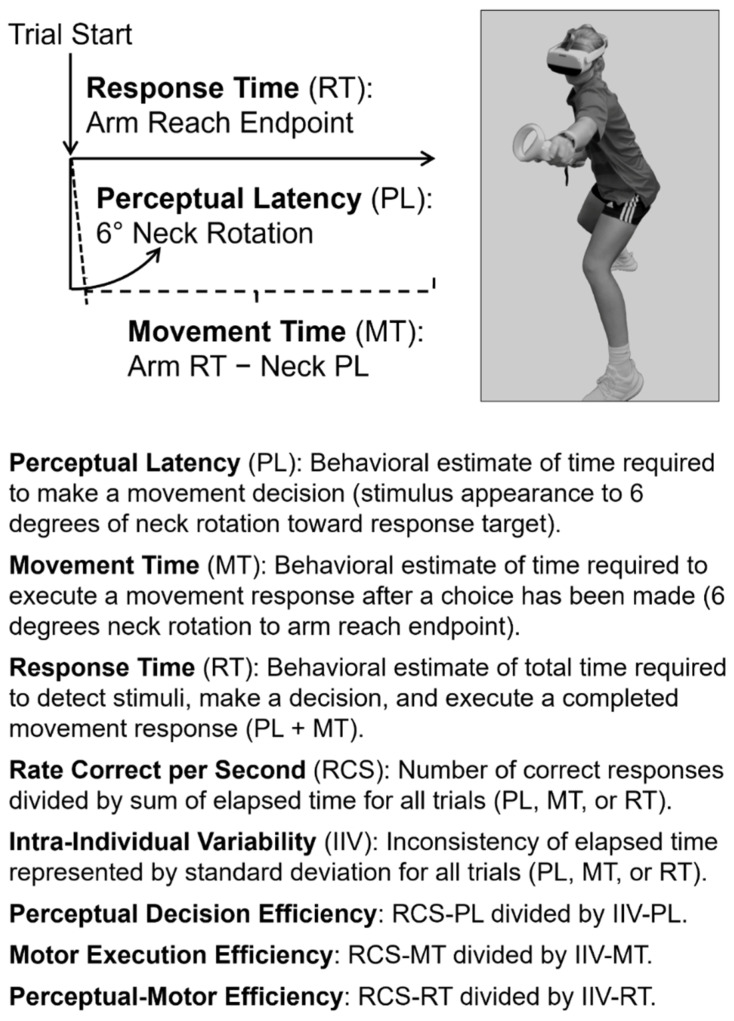
Depiction of time intervals derived from angular (neck rotation) and translatory (arm reaching) displacements of body segments toward a virtual target located beyond the peripheral field of view of the immersive virtual reality headset display, including definitions of performance metrics. The illustration photo is reproduced with permission from Ref. [59].

**Figure 3 brainsci-15-00721-f003:**
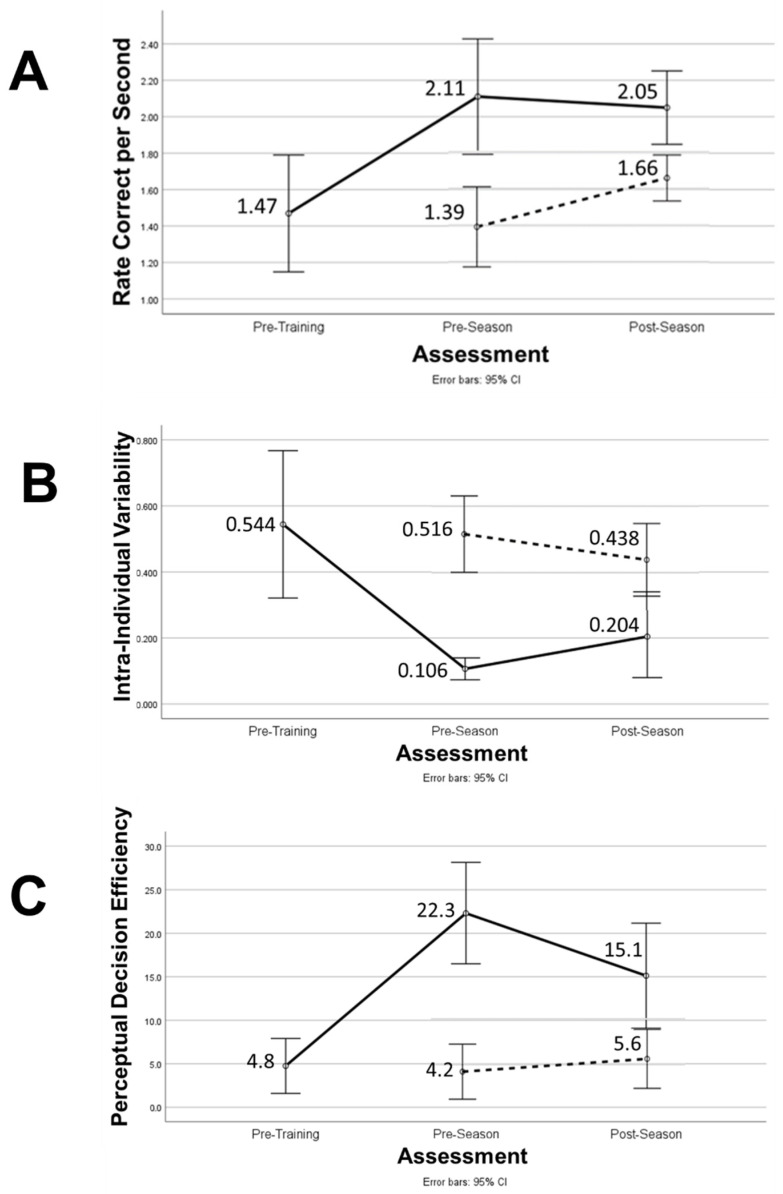
Pre- to post-training (i.e., pre-season) and post-season changes in (**A**) Rate Correct per Second, (**B**) Intra-Individual Variability, and (**C**) Perceptual Decision Efficiency for training group (solid lines) and pre- to post-season changes for comparison (no training) group (dashed lines). *Numerical* values are means derived from original (untransformed) data and error bars define corresponding 95% confidence intervals for the mean values. All pre- to post-training changes and group differences for pre-season and post-season were statistically significant (*p* < 0.05).

**Figure 4 brainsci-15-00721-f004:**
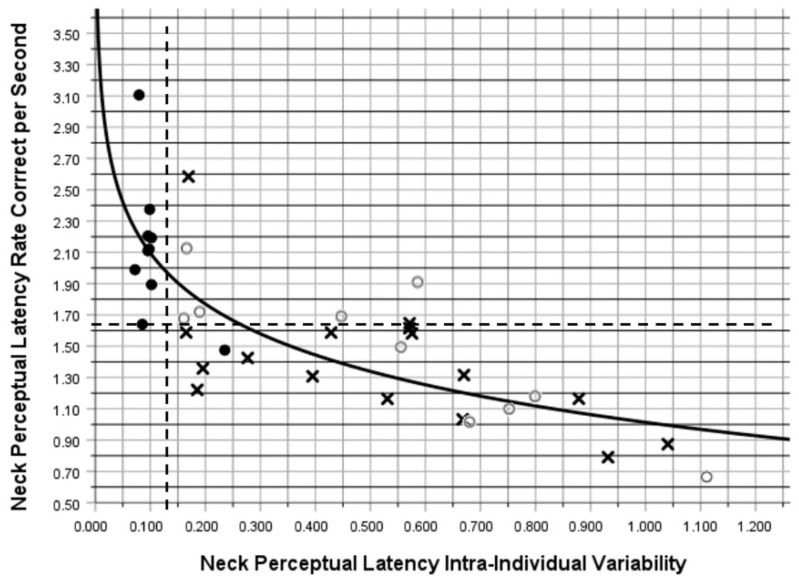
Inverse logarithmic correlation (curved solid line) between the Rate Correct per Second (RCS) and Intra-Individual Variability (IIV) performance metrics for neck perceptual latency at baseline (Spearman’s Rho = −0.795; *p* < 0.001). White circles identify baseline (pre-training) values for the 10 training group players and Xs identify baseline (pre-season) values for 16 untrained players. Dashed lines identify cut points for RCS (≤1.64) and IIV (≥0.134) that prospectively discriminate players who sustained an injury from those who remained uninjured for the entire season. Black circles identify post-training (pre-season) values for the 10 training group players.

**Figure 5 brainsci-15-00721-f005:**
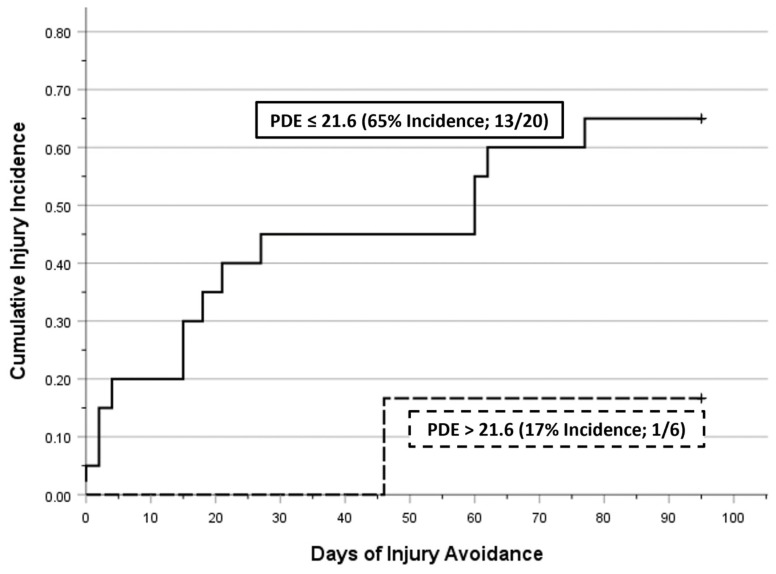
Kaplan–Meier depiction of musculoskeletal injury incidence (first injury event) across a 14-week period. A marginally significant difference (Mantel–Cox Log Rank *p* = 0.059) was documented between players categorized as low versus high performers on the basis of pre-season Perceptual Decision Efficiency (PDE ≤ 21.6 versus > 21.6).

**Table 1 brainsci-15-00721-t001:** Distribution Skew and Shapiro–Wilk Result (S-W *p*) and Pre- to Post-Training Change in Original (Untransformed) Metrics for Neck Perceptual Latency among Players in Training Group.

	Distribution Skew (S-W *p*)		Mean (Std Dev)		Difference	
Virtual Reality Metric	Pre-Training	Post-Training	Pre-Training	Post-Training	*p*	d
Rate Correct per Second	−0.327 (0.813)	1.013 (0.309)	1.47 (0.45)	2.11 (0.44)	0.012	1.00
Intra-Individual Variability	0.243 (0.484)	2.880 (<0.001)	0.544 (0.312)	0.106 (0.046)	0.002	1.34
Perceptual Decision Efficiency	1.025 (0.024)	0.169 (0.181)	4.75 (4.41)	22.30 (8.13)	<0.001	1.75

**Table 2 brainsci-15-00721-t002:** Distribution Skew and Shapiro–Wilk Result (S-W *p*) and Pre- to Post-Training Change in Natural Log-Transformed (Log_e_) Metrics for Neck Perceptual Latency among Players in Training Group.

	Distribution Skew (S-W *p*)		Geometric Mean (Log_e_)		Difference	
Virtual Reality Metric	Pre-Training	Post-Training	Pre-Training	Post-Training	*p*	d
Rate Correct per Second	−0.932 (0.367)	0.280 (0.692)	1.40 (0.34)	2.07 (0.73)	0.013	0.98
Intra-Individual Variability	−0.607 (0.003)	2.372 (<0.001)	0.448 (−0.80)	0.101 (−2.30)	<0.001	1.77
Perceptual Decision Efficiency	0.065 (0.530)	−1.885 (0.008)	3.12 (1.14)	20.60 (3.03)	<0.001	1.57

**Table 3 brainsci-15-00721-t003:** Distribution Skew and Shapiro–Wilk Result (S-W *p*) for Original (Untransformed) and Log-Transformed (Log_e_) Neck Perceptual Latency Metrics for Pre-Season and Post-Season Assessments.

	Distribution Skew (S-W *p*) Original Data		Distribution Skew (S-W *p*) Log_e_ Data	
Virtual Reality Metric	Pre-Season Skew (S-W *p*)	Post-Season Skew (S-W *p*)	Pre-Season Skew (S-W *p*)	Post-Season Skew (S-W *p*)
Rate Correct per Second	0.711 (0.389)	0.301 (0.849)	−0.114 (0.981)	−0.108 (0.934)
Intra-Individual Variability	0.892 (0.001)	0.786 (0.012)	0.128 (0.016)	−0.144 (0.032)
Perceptual Decision Efficiency	0.927 (0.001)	1.099 (<0.001)	−0.153 (0.082)	0.122 (0.128)

**Table 4 brainsci-15-00721-t004:** Pre- to Post-Season Change in Original (Untransformed) Immersive Virtual Reality Metrics for Neck Perceptual Latency among Players in Training Group (TG) versus Comparison Group (CG).

		Mean (Std Dev)		Group X Session Interaction		Group Difference	
Virtual Reality Metric	Group	Pre-Season	Post-Season	*p*	η_p_^2^	*p*	η_p_^2^
Perceptual Decision Efficiency	TG	22.31 (3.02)	12.71 (2.54)	0.017	0.216	<0.001	0.717
	CG	4.23 (1.11)	4.39 (1.48)				
Rate Correct per Second	TG	2.11 (0.72)	2.03 (0.70)	0.063	0.137	<0.001	0.510
	CG	1.39 (0.29)	1.64 (0.49)				
Intra-Individual Variability	TG	0.106 (0.046)	0.204 (0.174)	0.147	0.085	<0.001	0.503
	CG	0.516 (0.279)	0.438 (0.232)				

**Table 5 brainsci-15-00721-t005:** Pre- to Post-Season Change in Natural Log-Transformed (Log_e_) Immersive Virtual Reality Metrics for Neck Perceptual Latency among Players in Training Group (TG) versus Comparison Group (CG).

		Geometric Mean (Log_e_)		Group X Session Interaction		Group Difference	
Virtual Reality Metric	Group	Pre-Season	Post-Season	*p*	η_p_^2^	*p*	η_p_^2^
Perceptual Decision Efficiency	TG	20.60 (3.03)	12.71 (2.54)	0.038	0.168	<0.001	0.680
	CG	3.06 (1.12)	4.39 (1.48)				
Rate Correct per Second	TG	2.07 (0.73)	2.03 (0.70)	0.038	0.168	<0.001	0.511
	CG	1.34 (0.29)	1.64 (0.49)				
Intra-Individual Variability	TG	0.101 (−2.30)	0.160 (−1.83)	0.073	0.128	<0.001	0.653
	CG	0.437 (−0.83)	0.374 (−0.98)				

## Data Availability

The data presented in this study are available from the corresponding author upon institutional approval. The data are not publicly available, due to an institutional restriction on the release of data. A specific request from an individual who possesses research credentials must be reviewed and approved.

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
