# Peer review of "Perceptual Decision Efficiency Is Modifiable and Associated with Decreased Musculoskeletal Injury Risk Among Female College Soccer Players"

_brainsci, 2025, doi:10.3390/brainsci15070721_

Round 1
Reviewer 1 Report
Comments and Suggestions for Authors
- Relatively little research has addressed the central role of potentially modifiable cognitive processes in generating fast and effective motor responses to dynamic visual stimuli. The recommendation is to remove that paragraph from the abstract, as it requires a citation or citations.
The introduction is well-grounded from a theoretical perspective, and the authors have outlined the study’s two purposes precisely.
- After the last paragraph of the introduction, which outlines the two aims of the study, the authors must specify precisely what novel elements this study introduces and its place within the scientific literature.
- The recommendation is to specify that the study has complied with the Declaration of Helsinki (DoH)—Ethical Principles for Medical Research Involving Human Participants (1964) and its latest amendments adopted by the 75th General Assembly of the World Medical Association (WMA) in Finland on 19 October 2024.
- The recommendation is that sample size calculation, statistical power, or both should be included in the study.
- It is recommended to begin the discussion by outlining the two aims of the study and specifying whether or not they have been achieved.
- The primary limitation of this study was a lack of neurophysiological measurements of perceptual-motor processes that could provide direct evidence of correspondence with our behavioral metrics. Other limitations include its modest cohort number and lack of randomized assignment of participants to experimental and control groups, both of which precluded rigorous control of potentially confounding factors. Please include these paragraphs in Chapter 5. Limitations of the study
- Given the journal’s prestige, please remove or replace the following outdated bibliographical sources from the study: 22, 27, 37, 66, 70.
Moderate editing of the English language is needed.
Author Response
Comment 1: "Relatively little research has addressed the central role of potentially modifiable cognitive processes in generating fast and effective motor responses to dynamic visual stimuli." The recommendation is to remove that paragraph from the abstract, as it requires a citation or citations.
Response 1: The referenced text has been removed. The preceding sentence was modified to include “perceptual decision-making has only recently been recognized as a potentially important capability.”
Comment 2: After the last paragraph of the introduction, which outlines the two aims of the study, the authors must specify precisely what novel elements this study introduces and its place within the scientific literature.
Response 2: A sentence has been added that specifies the novel aspect of the study, which was correctly stated by a Reviewer 2 comment. “The novel aspect of this study is the derivation of a composite VR metric that is expected to be prospectively associated with the likelihood for musculoskeletal injury occurrence.”
Comment 3: The recommendation is to specify that the study has complied with the Declaration of Helsinki (DoH)—Ethical Principles for Medical Research Involving Human Participants (1964) and its latest amendments adopted by the 75th General Assembly of the World Medical Association (WMA) in Finland on 19 October 2024.
Response 3: The specified statement has been added to the first paragraph of the Methods section.
Comment 4: The recommendation is that sample size calculation, statistical power, or both should be included in the study
Response 4: Because the investigation was conceptualized as an exploratory cohort study, the number of participants was limited to the players included on the soccer team’s roster. As stated in line 156, the 10 players who were available to participate in the VR training sessions represented a convenience sample. In response to the request for statistical power calculations, post-hoc values for both the original (untransformed) and natural log-transformed data are reported in lines 254-256 for the Rate Correct per Second, Intra-Individual Variability, and Perceptual Decision Efficiency metrics.
Comment 5: It is recommended to begin the discussion by outlining the two aims of the study and specifying whether or not they have been achieved.
Response 5: The last paragraph of the Introduction section specified three purposes. A sentence was added at the beginning of the Discussion section: “Each of this study’s specified purposes were achieved.”
Comment 6: "The primary limitation of this study was a lack of neurophysiological measurements of perceptual-motor processes that could provide direct evidence of correspondence with our behavioral metrics. Other limitations include its modest cohort number and lack of randomized assignment of participants to experimental and control groups, both of which precluded rigorous control of potentially confounding factors." Please include these paragraphs in Chapter 5. Limitations of the study
Response 6: A ”4.2 Limitations” subheading has been added.
Comment 7: Given the journal’s prestige, please remove or replace the following outdated bibliographical sources from the study: 22, 27, 37, 66, 70.
Response 7: Although the five referenced studies are 17-19 years old, each one represents the earliest documentation of a key concept that has clearly been supported by the findings of subsequent research. Lo, et al. (2006) established the framework for subsequent development of the “drift-diffusion computational model of decision-making” that provides and explanation for the relationship we observed between the Rate Correct per Second and Intra-Individual Variability VR metrics. Bogacz, et al. (2006) and Ratcliff and McKoon (2008) were the first researchers to demonstrate the relevance of a two-alternative forced-choice task for application of the drift-diffusion model to human behavioral performance. Kelly, et al. (2008) provided the first evidence that linked across-trials behavioral variability to neuroimaging of brain network dynamics, and Churchland, et al. (2006) provided information that was foundational to our conceptualization of separable perceptual decision-making and motor execution phases of perceptual-motor responses to visual stimuli. Because our findings could not have been achieved without the foundational evidence provided by these studies, we respectfully suggest that they should be retained to recognize their contributions to the current state of knowledge on our chosen topic.
Reviewer 2 Report
Comments and Suggestions for Authors
The study investigates the impact of a virtual reality (VR) based perceptual-response training (PRT) on perceptual decision-making and its association with musculoskeletal injury risk in female collegiate soccer players. A total of 26 participants were divided into a training group (TG, 10 players) and a comparison group (CG, 16 players). The TG underwent a 10-session PRT protocol involving neck rotation, arm reach, and step-lunge responses to dynamic visual stimuli. The researchers defined a performance metric called perceptual decision efficiency (PDE), collected during VR training, to evaluate the impact of the VR based training. PDE is higher in the TG in both pre-season practice sessions and post-season period, and players with pre-season PDE>21.6 display longer injury avoidance days, suggesting that VR based PRT can improve perceptual decision-making and reduces injury likelihood, and PDE can be an effective measure to c provided good prospective for prospective prediction of musculoskeletal injury.
Overall, the novelty of the research is the PDE metric which synthesizes decision speed, accuracy and consistency and provide a single value to predict the likelihood of injury for women soccer players. The reason and procedure to calculate the metric and the experiment design are clearly described, and the result clearly showed the metric’s ability of distinguishing players with different injury likelihood. The finding of the study could be useful for injury prevention among female soccer players. I mainly have the following questions for the authors to consider:
- Since the 26 subjects are all from the same soccer team, one question is how their prior training could affect their outcome from VR training, as well as the performance from PDE metric. It is not clear for players that have different kinds of training and skill levels (for example, professional players and amateur players), whether similar results can still be achieved. The manuscript could include a discussion about this point.
- Related to the first point, the manuscript may want to discuss how individual subject’s prior experience level and performance can affect their performance and the PDE’s measure.
- While the authors acknowledged that this study lacks randomization of subjects between groups, the authors could discuss how the 16 CG and 10 TG group separation is determined
- It is interesting to know whether PDE can be calculated directly from real world behavior data, for example, from previous training of the players. Discuss about this point may enhance the applicability and impact of the research’s finding in the field
- The authors could include legend and statistical test results in the figure panels for better readability
Author Response
Comment 1: Since the 26 subjects are all from the same soccer team, one question is how their prior training could affect their outcome from VR training, as well as the performance from PDE metric. It is not clear for players that have different kinds of training and skill levels (for example, professional players and amateur players), whether similar results can still be achieved. The manuscript could include a discussion about this point.
Response 1: Content added to address these points in lines 433-440.
Comment 2: Related to the first point, the manuscript may want to discuss how individual subject’s prior experience level and performance can affect their performance and the PDE’s measure.
Response 2: Content added to address this points in lines 433-440.
Comment 3: While the authors acknowledged that this study lacks randomization of subjects between groups, the authors could discuss how the 16 CG and 10 TG group separation is determined.
Response 3: Because the investigation was conceptualized as an exploratory cohort study, the number of participants was limited to the players included on the soccer team’s roster. As stated in line 156, the 10 players who were available to participate in the VR training sessions represented a convenience sample. Most of them were incoming freshman and transfer players who were expected to have lesser performance capabilities than returning upper-class players who were not required to report early.
Comment 4: It is interesting to know whether PDE can be calculated directly from real world behavior data, for example, from previous training of the players. Discuss about this point may enhance the applicability and impact of the research’s finding in the field.
Response 4: With regard to real-world behavior data, a recommendation has been added to lines 471-473: “Future research might assess far-transfer of training benefit through comparison of post-training VR metrics to coaches’ ratings of on-field performance during competitive events.”
Comment 5: The authors could include legend and statistical test results in the figure panels for better readability.
Response 5: Content was added to the Figure 3 legend (lines 302-303): “All pre- to post-training changes and group differences for pre-season and post-season were statistically significant (p < 0.05).” Content was added to the Figure 5 legend (lines 315-316): “A marginally significant difference (Mantel-Cox Log Rank p = 0.059) was documented between players categorized as low versus high performers on the basis of pre-season Perceptual Decision Efficiency (PDE ≤ 21.6 versus > 21.6).” Additionally, labels have been added to Figure 5 to more clearly designate the meaning of the solid and dashed lines.